# Probing the Conformational States of Thimet Oligopeptidase in Solution

**DOI:** 10.3390/ijms23137297

**Published:** 2022-06-30

**Authors:** Marcelo F. M. Marcondes, Gabriel S. Santos, Fellipe Bronze, Mauricio F. M. Machado, Kátia R. Perez, Renske Hesselink, Marcel P. de Vries, Jaap Broos, Vitor Oliveira

**Affiliations:** 1Department of Biophysics, Universidade Federal de São Paulo, Rua Pedro de Toledo, 669, 7º Floor, São Paulo 04039-032, Brazil; marcelo.marcondes@unifesp.br (M.F.M.M.); ogsisan@gmail.com (G.S.S.); fellipe.bronze@gmail.com (F.B.); mauriciomachado@umc.br (M.F.M.M.); katia.perez@unifesp.br (K.R.P.); 2Department of Molecular Genetics, Groningen Biomolecular Science and Biotechnology Institute, University of Groningen, Nijenborgh 7, 9747 AG Groningen, The Netherlands; renske.hesselink@gmail.com; 3Interfaculty Mass Spectrometry Center, University of Groningen, A. Deusinglaan 1, 9713 AV Groningen, The Netherlands; marcel.de.vries@umcg.nl

**Keywords:** non-canonical amino acid, metallopeptidase, zinc-dependent peptidase, enzyme kinetics, peptidase family M3

## Abstract

Thimet oligopeptidase (TOP) is a metallopeptidase involved in the metabolism of oligopeptides inside and outside cells of various tissues. It has been proposed that substrate or inhibitor binding in the TOP active site induces a large hinge-bending movement leading to a closed structure, in which the bound ligand is enclosed. The main goal of the present work was to study this conformational change, and fluorescence techniques were used. Four active TOP mutants were created, each equipped with a single-Trp residue (fluorescence donor) and a *p*-nitro-phenylalanine (*p*NF) residue as fluorescence acceptor at opposite sides of the active site. *p*NF was biosynthetically incorporated with high efficiency using the amber codon suppression technology. Inhibitor binding induced shorter Donor-Acceptor (D-A) distances in all mutants, supporting the view that a hinge-like movement is operative in TOP. The activity of TOP is known to be dependent on the ionic strength of the assay buffer and D-A distances were measured at different ionic strengths. Interestingly, a correlation between the D-A distance and the catalytic activity of TOP was observed: the highest activities corresponded to the shortest D-A distances. In this study for the first time the hinge-bending motion of a metallopeptidase in solution could be studied, yielding insight about the position of the equilibrium between the open and closed conformation. This information will contribute to a more detailed understanding of the mode of action of these enzymes, including therapeutic targets like neurolysin and angiotensin-converting enzyme 2 (ACE2).

## 1. Introduction

Thimet oligopeptidase (TOP, EC 3.4.24.15) is a zinc ion dependent peptidase belonging to the large MA clan of metalopeptidases (M3 subfamily), which contains the characteristic HExxH motif [1]. This enzyme was first detected and isolated in 1983 from mouse brain extracts [2]. Studies using inhibitors suggest that TOP cleaves the biologically active peptides bradykinin, neurotensin, opioid peptides, angiotensin I and gonadotropin-releasing hormone [3] in the central nervous system and in other regions of the brain [4]. TOP can also process or degrade multiple peptides known as MHC Class I antigens [5]. These peptides are generated in the cytosol by the action of the proteasome, and the participation of TOP in the intracellular metabolism of these peptides, which makes it an integral part of the cellular immune system [6,7,8,9,10,11,12,13]. TOP is also involved in the intracellular processing of hemopressins, which can explain its importance for energy metabolism [11,14,15,16,17]. TOP features a broad substrate specificity [18,19] and its activity is known to be sensitive for changes in reaction conditions like a variation in temperature or ionic strength [20,21].

Metallopeptidases belonging to the MA clan feature a two-domain structure with the active site in a cleft at the domain interface. Binding of a substrate or ligand in the active site triggers a hinge-bending motion in the enzyme resulting in closing of the cleft [22,23,24,25,26,27,28]. The structure of TOP has been solved in the absence of a ligand and a deep groove is visible, the size and shape of which limits the access to peptides shorter than 17 residues [29]. Structures of related peptidases, crystallized in the absence and presence of a ligand, like dipeptidil carboxypeptidase (Dcp) [27], angiotensin-converting enzyme 2 (ACE-2) [28], and human neurolysin (NEL) [26] suggests that TOP also undergoes a hinge-bending motion induced by the binding of a substrate or an inhibitor at its active site. The large impact of His^600^ on TOP activity is another indication of the occurrence of this open–close molecular movement [30]. In the TOP crystal structure (open state) this His^600^ residue is too distant from the active site to be directly involved in substrate binding or hydrolysis. However, in the closed state of Dcp and ACE-2, the His^600^ counterparts His^505^ and His^601^, respectively, are close to the zinc ion in the active site [31] explaining the importance of this residue in the catalytic mechanism of this peptidase.

Hinge-bending motions in metallopeptidases have been reported for a number of enzymes [22,23,24,25,26,27,28] and is supported by crystallography data of the enzyme in absence and presence of a ligand, molecular dynamic simulations [32] and biochemical approaches [33]. However, to the best of our knowledge, studies investigating this conformational change in solution has not been reported to date. Information about the distribution of the metallopeptidases in the open and closed conformation and the impact of medium variations on this equilibrium is also not known.

In this paper a Förster resonance energy transfer (FRET) study is presented to collect this information for TOP. FRET can be used to study conformational changes in a protein, provided a donor-acceptor (D-A) couple suitable for probing relatively small changes in D-A distances is used. In this study we take advantage of the amber codon suppression labeling technology to introduce the unnatural amino acid *p*-nitro-phenylalanine (*p*NF) as acceptor in one of the two domains of TOP. Together with a single Trp residue as donor at the opposite domain, an attractive FRET couple was introduced to monitor hinge-bending motions in this peptidase. Our measurements provide direct experimental evidence that a hinge movement is operative in TOP. Distances between the probes were measured in buffers with different salt concentrations as the TOP activity is known to be sensitive for ionic strength [20,21]. Interestingly, a good correlation between D-A distances and catalytic activity was observed and these results are discussed in conjunction with previous studies about the mechanism of TOP.

Knowledge about parameters influencing the open/closed equilibrium of metallopeptidases is not only of interest for a better understanding of their catalytic mechanism. ACE2, a metallopeptidase that is structurally closely related to TOP is used as cellular receptor by corona viruses, causing SARS and COVID-19 [34]. A viral protein, spike glycoprotein, binds with nanomolar affinity to ACE2, a key step for viral invasion into the host [34,35]. Recently, the 3D structures of spike glycoprotein in complex with ACE2 were solved and in all structures the spike glycoprotein is in complex with the open conformation of ACE2 [34,35,36], suggesting it shows the highest affinity for this state. The significance of this study for the complex formation between spike glycoprotein and ACE2 is discussed.

## 2. Results and Discussions

FRET was used to study conformational changes in TOP induced by variations in ionic strength of the buffer or by inhibitor binding. Preferably, the donor and acceptor probes are compact in size and have short linkers connecting them with the protein backbone, in this way yielding sharply defined FRET efficiencies. The hinge-bending movement can be most sensitively probed if the D-A pair is positioned at opposite sides of the central groove. With this in mind, we selected two amino acids as probes, namely Trp as donor and the unnatural amino acid *para*-nitrophenylalanine (*p*NF) as acceptor. The later was introduced using the amber codon suppression technique [37,38,39]. In this way both probes were bio-synthetically incorporated during *E. coli* expression. We have also explored the engineering of cysteine residues in TOP with the aim being to introduce a spectroscopic probe, near the central groove. This approach was abandoned as most TOP activity was lost during the in vitro labeling. TOP has 14 Cys residues (all free—SH) in its structure, and it is well-known that some of them are very reactive and also that the chemical modification of such Cys residues blocks TOP activity [40,41].

Below, the design of suitable TOP mutants is presented, followed by their characterization using mass spectrometry, activity assays and fluorescence measurements.

### 2.1. Single-Trp TOP Mutants

WT TOP from rat has 6 Trp residues and mutants were created to investigate the importance of each Trp residue on the enzyme expression level and enzyme activity. The following TOP mutants were investigated: W124F; W355F; W390F; W614F; and W511F/W513F. Wild type TOP and these five mutants were expressed in *E. coli* and after cell disruption and centrifugation of the suspension, the expression levels were evaluated using SDS PAGE gel (Appendix A (Panel B)). The results clearly show Trp390 is important for the expression of TOP as soluble protein and for the enzymatic activity of TOP (Appendix A). These experiments indicate Trp390 as the most suitable donor position for the FRET experiments.

Based on these observations, we pursued with the construction of a TOP mutant with all Trp residues replaced by Phe residues (TOP^-Trp^) and the single-Trp TOP mutant, TOP^-Trp^W390, in which all Trp residues were replaced by Phe except Trp390. The expression of these two mutants was verified and compared with WT TOP (Appendix A (Panel C)). The TOP^-Trp^ mutant showed a relatively good expression level, however, almost all protein was in the insoluble fraction as observed for the TOP W390F mutant (Appendix A (Panel B)). On the other hand, the single-Trp mutant (TOP^-Trp^W390) showed an expression level comparable with WT TOP and the expressed protein was in the soluble fraction (Appendix A (Panels C and D)). Analysis of the specific activity with QFS as substrate (a commonly used TOP substrate) showed that mutant TOP^-Trp^W390 was active, albeit significantly reduced when compared to WT TOP (Table 1). Therefore, this single-Trp mutant (TOP^-Trp^W390) was selected as a basis for a set of TOP mutants with Trp390 as fluorescence donor probe and pNF as acceptor.

### 2.2. Double Labeled TOP Mutants (Trp/pNF)

To get an impression of the changes in TOP structure due to the hinge movement, the closed TOP structure was modeled starting with the crystallographic “open” TOP structure (Figure 1) (modeling details are in the Appendix A). After superimposing both structures, potential positions for *p*NF were chosen in domain A (acceptor) facing domain D (donor), which harbors Trp390 (Figure 1C and Figure 2) considering the 15.8 Å Förster distance (*R*_0_) of the Trp-*p*NF couple. Aromatic residues or bulky hydrophobic residues were preferentially chosen to be replaced by the *p*NF residue. Using this approach, four different residues (Phe178, Leu182, Leu215 and Tyr224) were selected as shown in Figure 2. Attempts to introduce *p*NF at residue positions in the (A) acceptor domain spatially closer to W390 failed as the resultant TOP mutants were inactive (unpublished results).

Four new constructs were made, each with an amber codon encoding for residue positions 178, 182, 215 or 224, respectively, yielding constructs TOP^-Trp^W390/F178TAG; TOP^-Trp^W390/L182TAG; TOP^-Trp^W390/L215TAG and TOP^-Trp^W390/Y224TAG. The TAG (UAG) amber stop codon makes possible the site selective introduction of *p*NF by *E. coli* when the expression host is equipped with the orthogonal tRNA^UAG^—tRNA synthetase pair evolved for the incorporation of *p*NF. We developed such a pair, guided by a similar system reported by Schultz et al. [39] (see Section 3.3).

In gel tryptic digestion experiments were performed followed by mass spectrometry (LC/MS) to establish the *p*NF incorporation efficiencies. Residue position 224 is expected in a nonamer tryptic peptide and this peptide was found next to a decamer and a 13-mer peptide harboring residue 224. For mutant TOP^-Trp^ W390/Y224 *p*NF, 96.2% of the peak intensity was of *p*NF labeled peptides, while 1.1% contained a Phe at position 224. Residue position 182 is predicted in a 13-mer tryptic peptide and only this 13 mer was found when analyzing mutant TOP^-Trp^W390/L182*p*NF. The unlabeled 13-mer was not detected indicating complete *p*NF labeling of residue position 182. Residue position 215 is predicted in a tryptic dipeptide. *p*NF-labeled peptides were detected, but the low mass of this dipeptide, combined with many tryptic cleavage sites close to residue 215, precluded estimating the *p*NF incorporation efficiency in mutant TOP^-Trp^W390/L215*p*NF. Finally, residue position 178 is expected in a decamer trypic peptide, however this residue position could not be detected in a tryptic peptide when analyzing mutant TOP^-Trp^W390/F178*p*NF or the other three mutants. Together the protocol used to express *p*NF- labeled TOP mutants results in a very efficient site-specific labeling of this metallopeptidase by the FRET acceptor *p*NF. This result is important as a high incorporation efficiency ensures that essentially all W390 residues in the sample can interact with an acceptor, validating the use of integrated donor emission intensities for calculating the D-A distances.

The specific activity of all isolated mutants was checked using the QFS substrate and the *K*_i_ value was determined for JA-2 (Appendix A), a well-known linear-competitive TOP inhibitor (Table 1) [43].

TOP^-Trp^W390 present a *k*_cat_/*K*_M_ 27 times lower than for the WT protein, showing the impact of 5 Trp→Phe replacements on TOP activity. A 27-times lower *k*_cat_/*K*_M_ is significant, but given the high catalytic power of proteases to cleave peptide bonds, the mutant can still be regarded as a very good catalyst. For example, for the metallopeptidase carboxypeptidase B, a 1.3 × 10^13^ times increase in peptide hydrolysis rate has been reported compared to the uncatalyzed hydrolysis rate [44].

Similar activities were observed for TOP^-Trp^W390, TOP^-Trp^W390/L215*p*NF and TOP^-Trp^W390/Y224*p*NF. For the mutants TOP^-Trp^W390/F178*p*NF and TOP^-Trp^W390/L182*p*NF a significant lower *k*_cat_/*K*_M_ value was measured compared with TOP^-Trp^W390.

### 2.3. Concentration of the Purified TOP Mutants

FRET efficiencies (E) are calculated by comparing the integrated emission intensities of the donor only spectrum and of the D-A pair, using equimolar concentrations. The calculated E values therefore heavily rely on the experimentally determined protein concentrations and two independent methods were used to measure the protein concentrations. First a Bradford assay was used, and the obtained concentrations were checked using a fluorescence approach. In this latter approach the TOP proteins were denatured using guanidinium chloride (GdmCl) as this allows a comparison of the Trp emission intensities as no FRET is expected to take place when the proteins are denatured. In Appendix AA,B emission spectra are presented for TOP^-Trp^W390 and the double labeled TOP^-Trp^W390/Y224*p*NF mutant at different GdmCl concentrations. Addition of 1 M GdmCl does not induce changes in the emission spectra suggesting the structure of the proteins are conserved. At 2 M GdmCl the emission spectra start to shift to the red, indicating that the proteins start to lose their structural integrity, and at higher GdmCl concentrations the emission spectra are further shifted to the red, indicating that the proteins are denatured. The integrated emission intensity for the double labeled mutant (TOP^-Trp^W390/Y224*p*NF), after correction for dilution, decreases upon increasing the concentration of GdmCl. Apparently, the quantum yield of the Trp residue is decreased by adding GdmCl and the same phenomena was observed for the single-Trp mutant TOP^-Trp^W390. At denaturing condition (6 M of GdmCl), the integrated fluorescence area of both TOP^-Trp^W390 and the double labeled TOP^-Trp^W390/Y224*p*NF mutants become almost the same (only 1% difference), confirming that equimolar concentrations of proteins have been used in these denaturation experiments.

### 2.4. Donor-Acceptor Distances in TOP Mutants

Based on the solved “open” TOP crystal structure, the D-A distance in all mutants are >>*R*_0_ and should correspond to a very low E (E < 0.02) (Table 2). However, a detectible E (E = 0.07–0.27—Table 2) was observed, corresponding to distances being 12–17 Å shorter than the expected distances. We note that the expected distances presented in Table 2 are based on Cβ-Cβ distances of each D-A couple in the crystal structure of WT TOP and the actual D-A distances could deviate due to a different rotamer distribution. Differences in the overlap integral *J*, Quantum yield *Q*, or orientation factor *κ*^2^ between the open and closed state could also cause deviations from the expected D-A distances, which are calculated using *R*_0_ = 15.8 Å. Based on experiments with double labeled mutants and TOP^-Trp^W390 in absence and presence of inhibitor we can exclude that variations in *J* and *Q* are responsible for the large D-A distance deviations (Appendix A). The impact of *κ*^2^ is more difficult to evaluate but B factors for the acceptor residue positions (Appendix A) [29] indicate rotational mobility of the side chains is expected for the four acceptor positions used in this study, especially for the L215 and Y224 residues. A time-resolved fluorescence anisotropy analysis of donor position W390 shows side chain rotation within the protein structure (Appendix A). This and the low intrinsic anisotropy value of Trp (r_0_ = 0.25) [45] when excited at 295 nm, due to two overlapping transitions which excitation transition moments are almost perpendicular with in respect to each other, support the use of a *κ*^2^ value of 2/3 in this study, although some deviation in this *κ*^2^ value cannot be ruled out.

The significantly shorter experimental D-A distances than expected supports the notion that TOP exists in solution as a mixture of open, closed and/or intermediate conformation(s) between the open and closed state. The consensus view in the metallopeptidase field is that in the absence of a ligand (substrate, inhibitor) the MP is in the open state, while interaction with a ligand in the active side brings it in the closed conformation. This view is essentially based on protein crystallography and molecular dynamics studies of metallopeptidases in absence and presence of ligand. An exception is known, namely the testis angiotensin converting enzyme, the structure of which (Protein Data Bank code 1O8A) shows this enzyme adopts a “closed” conformation in the absence of a ligand in its active site [46]. This ligand free tACE1 structure is almost identical to an inhibitor-bound structure [46,47]. Although tACE1 does not show a high sequence identity with TOP, this peptidase has the same overall fold as reported for TOP, including a centrally located deep groove. In summary, the FRET data for the TOP mutants in the absence of ligand correspond to significant shorter D-A distances as in the crystal structure of WT TOP in the open state. This is likely due to conformational heterogeneity reflecting the presence of a significant fraction of TOP in the closed state and possibly in states in between the open and closed states. To the best of our knowledge, this is the first-time experimental data about the open/closed states of a metalloprotein, belonging to the MA clan, in solution is presented.

### 2.5. Conformational Changes in TOP Mutants upon JA-2 Binding

For the assays in the presence of JA-2, saturating concentrations of this inhibitor were used ([JA-2] ≥ 10 × *K*_i_; Table 1), resulting in more than 90% of the enzyme in a ligand bound state. For all double-labeled mutants, the addition of JA-2 induced higher E values (Table 2), indicating that the interaction between TOP and this ligand moves the equilibrium to the closed state. The measured and modeled D-A distances in the presence of JA-2 for the TOP^-Trp^ W390/F178*p*NF and TOP^-Trp^ W390/L182*pNF* mutants show the same and a 2 Å difference, respectively, while the deviation is 8–9 Å, for the TOP^-Trp^W390/Y224*p*NF and TOP^-Trp^W390/L215*p*NF mutants. Residues Tyr224 and Leu215 are both in a more flexible region as represented in the temperature B-factor colored structure (Appendix A), than positions 178 and 182 and the deviation might be related to this. The calculated reductions in D-A distances induced by JA-2 binding are small, only 1–2 Å (Table 2), while modeling suggests a change of 8–11 Å (Figure 2). This discrepancy can be explained by a large fraction of the mutant already in the closed state in this buffer. The percentage of the TOP mutant in either open or closed conformation can be calculated if one assumes a simple model, in which the enzyme can only exist in either the open or closed conformation. According to this model, 65% to 90% of the investigated TOP mutants are present in the closed state. For example, for mutant TOP^-Trp^W390/Y224*p*NF in the absence of JA-2, 90% is calculated to be in the closed state (Table 2, last column) and only the remaining 10% in the open state can enhance E upon JA-2 binding. As only one out ten mutant molecules can enhance E upon JA-2 binding, the calculated shortening in D-A distance (1 Å) is a strongly underestimated value.

We note that the measured changes in E upon binding of JA-2 are also relatively small (≤0.07) as the D-A distances in absence and presence of JA-2 are all larger than 15.8 Å, the Förster distance of the Trp-*p*NF couple. Despite the small changes in E induced by JA-2 binding, for all mutants an increase in E was measured upon JA-2 binding and combined with the small errors in these measurements (Table 2, Figure 3), we conclude that upon JA-2 binding the two domains in the TOP structure move towards each other. Together, binding of JA-2 induces shorter D-A distances in all four mutants, an observation in line with the presence of a hinge-bending movement in TOP.

### 2.6. Effects of NaCl on the Structure-Activity Relationship of TOP

Previous studies showed that the TOP activity can be changed an order of magnitude when the ionic strength in the assay buffer is varied [20,21] and this prompted us to investigate if TOP activity is related with structural changes in the enzyme, as expressed in different D-A distances (i.e., different E—FRET efficiencies).

First, kinetic assays were made to measure the impact of NaCl (0–2.5 M) on the activity of the single Trp mutant TOP^-Trp^W390 (Table 3). The impact of the 5 Trp to Phe mutations in this enzyme on the activity at different NaCl concentrations is not known and therefore WT TOP was included in this study. Upon increasing the concentration of NaCl from zero to 0.5 M, the TOP catalytic efficiency (*k*_cat_/*K*_M_) decreases ~2 fold for both WT TOP and the TOP^-Trp^W390 mutant, while a further increase in NaCl concentration from 0.5 M to 2.5 M NaCl increases the *k*_cat_/*K*_M_~7–12 fold (Table 3). The results presented in Table 3 demonstrate that, despite the large difference in absolute activity, the activity of both enzymes’ response is similar when the NaCl concentration is changed. This supports the use of TOP^-Trp^W390 constructs as a model protein to investigate the impact of NaCl on the open/close equilibrium of TOP.

Second, the fluorescent properties of the single-Trp mutant (TOP^-Trp^W390) and the double-labeled mutant TOP^-Trp^W390/Y224*p*NF were analyzed in the assay buffer containing 0–2.5 M NaCl and the results are presented in Table 4. Gradually increasing the NaCl concentration from zero to 0.5 M decreases E from 0.27 to 0.11. These E values correspond to 90% and 34%, respectively, of TOP in the closed state. Thus, TOP is changing towards a more open conformation when the salt concentration is increased. However, at 2.5 M NaCl, the E value becomes 0.35, corresponding to all TOP in the closed state. Interestingly, the kinetic and FRET data show a correlation, the higher the activity, the higher the fraction of TOP in the closed state. Intuitively, one expects TOP in the open state being most active and we can only speculate why TOP in the closed conformation is more active and if it is of biological relevance for the in vivo function of TOP. Possibly, during evolution TOP has prevailed in the closed conformation as it is more stable than in the open state and the formation of a productive enzyme substrate complex is best achieved when the substrate initially interacts with TOP in a closed conformation. More work is needed to investigate the impact of other salts and substrates on TOP activity and the open/closed equilibrium. Together, the activity assays and FRET data indicate that variation in the NaCl concentration in the assay buffer affects both the balance between open and closed conformations of TOP, as well as its activity.

### 2.7. Impact of the Open/Closed Equilibrium on the Interactions of Metallopeptidases with Other Proteins

It has been reported that TOP can interact in vivo with calmodulin or with 14.3.3epsilon in this way, stimulating the secretion of TOP in the cell [17,48]. It might be of interest to investigate if this interaction is influenced by the position of the open/closed equilibrium of TOP.

For the structurally related metallopeptidase human ACE2, crystal structures for the open and closed states have been solved [28]. ACE2 is used as a cellular receptor by coronaviruses, causing SARS and COVID-19. The spike glycoprotein from coronaviruses binds to ACE2 with high affinity (*K_d_* at the low nM range) and the two most important attachment sites of this viral protein at ACE2 (called hotspots) are two salt bridges located at the ACE2 N-terminal domain [34,35,36]. This high affinity is believed to make the coronaviruses very contagious as binding of the spike protein to ACE2 is a key step in the infection process [49]. X-ray and Cryo-Electron microscopy were recently used successfully to solve the structures of ACE2 in complex with spike proteins from different corona viruses, SARS-CoV1 and SARS-CoV2. All these structures revealed the spike protein binds to ACE2 in the open conformation [34,35,36,50,51]. This suggest the highest affinity is obtained between the two proteins when ACE2 is in the open conformation. Currently the affinity of the spike protein for ACE2 in the closed conformation is not known and no information is available about the open/closed equilibrium of ACE2. Transferring ACE2 to a closed conformation could become a strategy in the fight against coronavirus infections. Our work with TOP shows this is possible without the need of an inhibitor, in this way keeping the enzyme activity intact.

## 3. Materials and Methods

### 3.1. Site-Directed Mutagenesis

A Quick-change site-directed mutagenesis kit (Stratagene, La Jolla, CA, USA) was used to introduce specific point mutations in wild type TOP from rat (UniProtKB—P24155), as previously described [52]. The point mutations were all confirmed by DNA sequencing.

### 3.2. Protein Expression (Wild-Type and Non pNF Labeled TOP Mutants)

Wild type and mutant proteins were expressed in *Escherichia coli* BL21 (DE3) using the plasmid pET26b containing cDNA encoding the desired protein. Typical expression conditions were: Pre-culture—10 mL of Luria-broth (LB) medium containing kanamycin (50 µg/mL) shaken at 150 rpm overnight at 37 °C; Culture—pre-culture added to 1 L of fresh LB medium containing kanamycin (50 µg/mL) shaken at 150 rpm and at 30 °C until the culture density reached 0.4 OD_550_, when the temperature was lowered to 20 °C. Growth was continued till 0.6–0.7 OD_550_ when 0.5 mM isopropyl β-D-1-thiogalactopyranoside (IPTG) was added. After 14–16 h, the bacterial cells were harvested by centrifugation at 3200× *g*, for 10 min, at 4 °C. The pellet was stored at −70 °C.

### 3.3. TOP Labeling with p-Nitro-Phenylalanine Residue

The amber codon suppression methodology developed by P. Schlutz et al. [53,54,55] was used to incorporate biosynthetically *p*-nitro-phenylalanine (*p*NF) residues [39] was built with some modifications. The genes of the orthogonal pair tRNA^UAG^ and tRNA aminoacyl-synthetase, evolved to biosynthetically incorporate *p*NF (tRNA_RS_*p*NF) in *E. coli* were synthetized (Epoch Life Sciences, Missouri, TX, USA). The synthetic genes were cloned in plasmid pACYC184 as follows: the tRNA^UAG^ coding sequence, including lpp promoter and rrnC terminator regions, were cloned between the Bsu36I and PflFI restriction sites and the T7 promoter + tRNA_RS_*p*NF coding sequence + terminator was inserted between BamHI and SalI restriction sites of the same plasmid.

*Escherichia coli* BL21 (DE3), co-transfected with plasmid pET26b containing cDNA encoding the desired TOP mutant and the pACYC184 plasmid carrying the genes of the orthogonal tRNA_RS_*p*NF pair, were used for the expression of TOP labeled with *p*NF residues. The expression conditions were the same as described above for WT TOP, including the pre-culture in LB medium but for the expression step the LB medium was replaced by M9 medium [56] supplemented with 1 mM *p*NF.

### 3.4. Purification of TOP Proteins

Bacterial cells were suspended in 20 mL of binding buffer (50 mM NaH_2_PO_4_, 300 mM NaCl, 20 mM imidazole, 1 mM DTT, pH 7.4) and submitted to a French press procedure (2 times at 10,000 psi). The cell lysate was treated using 2.4 units/mL of Benzonase^®^ Nuclease, 1 mM of MgSO_4_, and incubated at room temperature for 15 min. The cell lysate was centrifuged at 27,000× *g* for 20 min and the supernatant was incubated under agitation during 30 min at 8 °C with 1 mL Ni-sepharose FF resin (GE Healthcare, Chicago, IL, USA), pre-equilibrated with binding buffer. The slurry was poured in a column and the resin washed with 8 mL of binding buffer (4 × 2 mL). The protein was eluted with 4 × 1 mL elution buffer (20 mM NaH_2_PO_4_, pH 7.4, 300 mM NaCl, 150 mM Imidazol, 1 mM DTT). The fractions containing WT TOP or mutant were desalted using a PD10 column (GE Healthcare) equilibrated with TB buffer (Tris 50 mM, 1 mM DTT pH 7.4). Desalted samples were loaded at a flow rate of 1.0 mL/min on a 1 mL resource Q column (GE Healthcare) equilibrated with TB buffer. After an initial washing step using 6 mL of TB buffer, the elution of protein was performed using a linear gradient of 20 mL TB buffer and 20 mL TBS buffer (50 mM Tris-HCl, pH 7.4, 500 mM NaCl 1 mM DTT). TOP eluted between 80–200 mM NaCl. The fractions were analyzed using an SDS-PAGE gel and fractions containing pure protein were concentrated and dialyzed using an Amicon filtration unit (Millipore Corp., Burlington, MA, USA) equipped with a 30 kDa exclusion membrane. The recovered protein was stored at −70 °C.

### 3.5. Protein Concentration

Protein concentrations were determined as described by Gill and von Hippel [57], while those for spectroscopy experiments were determined by the Bradford dye-binding assay [58] using bovine serum albumin as standard.

### 3.6. Mass Spectrometry

Online chromatography of the tryptic peptides was performed with the Ultimate 3000 nano-HPLC system (Thermo Fisher Scientific, Waltham, MA, USA) coupled online to a Q-Exactive-Plus mass spectrometer with a NanoFlex source (Thermo Fisher Scientific) equipped with a stainless-steel emitter. Tryptic digests were loaded onto a 5 mm × 300 μm i.d. trapping micro column packed with PepMAP100 5 μm particles (Dionex, Sunnyvale, CA, USA) in 0.1% FA at the flow rate of 20 μL/min. After loading and washing for 3 min, peptides were forward-flush eluted onto a 15 cm × 75 μm i.d. nanocolumn, packed with Acclaim C18 PepMAP100 2 μm particles (Dionex). The following mobile phase gradient was delivered at the flow rate of 300 nL/min: 2–50% of solvent B in 60 min; 50–90% B in 1 min; 90% B during 14 min, and back to 2% B in 1 min and held at 2% A for 15 min. Solvent A was 100:0 H_2_O/acetonitrile (*v*/*v*) with 0.1% formic acid and solvent B was 0:100 H_2_O/acetonitrile (*v*/*v*) with 0.1% formic acid. MS data were acquired using a data-dependent top-10 method dynamically choosing the most abundant not-yet-sequenced precursor ions from the survey scans (300–1650 Th) with a dynamic exclusion of 20 s. Sequencing was performed via higher energy collisional dissociation fragmentation with a target value of 1 × 10^5^ ions determined with predictive automatic gain control. Isolation of precursors was performed with a window of 2.0 *m*/*z*. Survey scans were acquired at a resolution of 70,000 at *m*/*z* 200. Resolution for HCD spectra was set to 17,500 at *m*/*z* 200 with a maximum ion injection time of 50 ms. The normalized collision energy was set at 27. Furthermore, the S-lens RF level was set at 60 and the capillary temperature was set at 250 °C. Precursor ions with single and unassigned charge states were excluded from fragmentation selection.

The LCMS data was searched using PeaksStudioX against a user database containing the protein of interest and a contaminant database. Data was searched with a parent mass accuracy of 20 ppm and a fragment mass error tolerance of 0.04 Da. Trypsin as enzyme and 2 missed cleavages were allowed. The following variable modifications were allowed: −15.99 (Y→F substitution), 28.99 Da (Y→ F-NO_2_), +44.99 Da (F→F-NO_2_), 33.98 (L→F), 49.97 Da (L→Y), 78.96 (L→-F-NO_2_) and oxidation of methionine. Carbamidomethylation of cysteine was used as fixed modification. For the calculation of the *p*NF incorporation efficiency, the peak intensities of the identified modified and unmodified peptides were used.

### 3.7. Kinetic Assays

WT TOP and mutant activities were monitored spectrofluorometrically in a Shimadzu RF-5301PC spectrofluorometer using the FRET peptide QFS (Mca-Pro-Leu-Gly-Pro-D-Lys (Dnp)-OH) as substrate [59], with excitation and emission wavelengths of 320 and 420 nm, respectively. TOP and its mutants were pre-activated by incubation with 0.5 mM DTT for 5 min at 37 °C. The concentration of the peptide solutions was determined by measuring the absorption of the 2,4-dinitrophenyl group at 365 nm (ε = 17,300 M^−1^cm^−1^). The kinetic parameters of peptide hydrolysis were determined at 37 °C in 50 mM Tris-HCl buffer, pH 7.4, containing 100 mM NaCl. A standard cuvette (1 cm pathlength) containing 1 mL of substrate solution was placed in a thermostatically controlled cell holder for 5 min before the addition of enzyme. The reaction was monitored continuously via the fluorescence of the released product. The rate of increase in fluorescence was converted into moles of substrate hydrolyzed per second based on the fluorescence curves of standard peptide solutions before and after total enzymatic hydrolysis. The enzyme concentration for initial rate determinations was chosen so that <5% of the substrate was hydrolyzed. An empirical equation was used to correct for the inner-filter effect [60], and the kinetic parameters were calculated according to Wilkinson [61] and by using Eadie–Hofstee plots. Grafit (version 5.0; Erithacus Software) was used to fit the data.

### 3.8. Determination of Inhibition Parameters

Inhibition constants were determined using the TOP inhibitor JA-2 (*N*-[1-(R,S)-carboxy-3-phenylpropyl]Ala-Aib-Tyr-p-aminobenzoate) [43]. The constants were determined in a continuous assay using the FRET peptide QFS as substrate. The equation used to calculate the *K_i_* values was *K_i_* = *K_i,app_*/(1 + [S]/*K_M_*), where [S] = molar concentration of the substrate, *K_M_* = Michaelis constant, and *K_i,app_* = apparent inhibition constant. *K_i,app_* was calculated using the equation *V_o_*/*V_i_* = 1 + [I]/*K_i,app_*, where *V_o_* = velocity of hydrolysis without the inhibitor, *V_i_* = velocity of hydrolysis in the presence of the inhibitor, and [I] = molar concentration of the inhibitor. A plot of (*V_o_*/*V_i_*) − 1 vs. [I] yields a slope of 1/*K_i,app_*.

### 3.9. Protein Fluorescence Assays

FRET studies were conducted using a Fluorlog 322 fluorometer (Jobin Yvon, Edison, NJ, USA) at 25 °C using an excitation wavelength of 295 nm, and excitation and emission slits at 1.25 nm and 5.0 nm, respectively, conditions preventing photobleaching of the samples. Then, 0.5 µM TOP protein was analyzed in a quartz cuvet using conditions as described in the Section 2. Emission spectra were recorded between 305–450 nm and the spectra were corrected for buffer and instrument response and the intensities integrated. For the assays using the JA-2 compound, 13.4 µM of inhibitor was used, under these conditions 92% of enzyme will be complexed, considering that the highest *Ki* value obtained was 1.2 µM (Table 1).

The distance (*r*) between the 2 chromophores were calculated using Equations (1) and (2).
(1)$$E=\frac{1}{1+{\left(\frac{r}{R_{0}}\right)}^{6}}$$
(2)$$E=1-\frac{F_{DA}}{F_{D}}$$
where *E* is the energy transfer efficiency between the donor–acceptor chromophores, *R*_0_ the Förster distance, *F_DA_* the fluorescence intensity in the presence of the acceptor and *F_D_* the fluorescence intensity in the absence of the acceptor (spectrum of TOP^-Trp^W390). The *R*_0_ distance was calculated using software provided in [62]. The following constants were used: Dipole orientation factor, *κ*^2^ = 0.66; Refractive index medium, *η* = 1.4; Extinction coefficient of *p*NF 9700 M^−1^cm^−1^ at 275 nm [63]. The quantum yield of the donor, TOP^-Trp^W390 is ϕ = 0.13. It was determined in 50 mM Tris pH = 7.4 + 100 mM NaCl using *N*-acetyl-tryptophanamide (ϕ = 0.14) as reference.

## 4. Conclusions

In this work experimental data is presented about the TOP structure in solution in unliganded and liganded states. At most assay conditions, the results indicate most TOP molecules are in a closed conformation, even in the absence of active site ligands. However, the existence of other intermediate states between open or closed cannot be excluded. Binding of the specific active site inhibitor JA-2 shortens the D-A distances in all four mutants investigated, providing for the first-time experimental support that a hinge bending movement takes place in this enzyme upon ligand binding. The open/closed equilibrium and the TOP activity was found to be sensitive to the ionic strength in the medium and the similar structure of related metallopeptidases, including the therapeutic targets neurolysin, ACE1 and ACE2, making it likely that their open/closed equilibria and activity are also sensitive to ionic strength.

The FRET-based method developed for studying the conformational states of TOP in solution makes it possible to analyze new aspects of the TOP mechanism. For example, pre-steady experiments can elucidate if the transition from one conformational state to the other is rate limiting and what the impact of ligand structure and buffer conditions are on these kinetics. The ability to monitor the open/closed equilibrium gave new insight into the mechanism of TOP and is a new instrument to study the mechanisms of metallopeptidases.

## Figures and Tables

**Figure 1 ijms-23-07297-f001:**
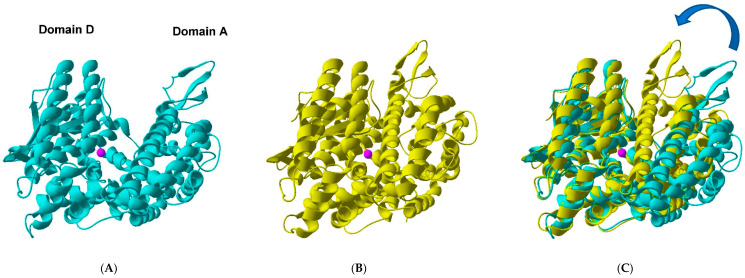
Open versus closed TOP. (**A**) Ribbon representation of the “open” TOP crystal structure (PDB 1S4B). (**B**) Ribbon representation of the modelled “closed” TOP structure. The model was based on the recently published structure of human neurolysin E475Q mutant in complex with neurotensin peptide products (PDB 5LUZ). (**C**) Overlay of the two TOP molecules fixing the zinc binding site HExxH + E + Zn^2+^ (pink sphere). The arrow highlights the relative change in position of domains D(onor) and A(cceptor) that should occur considering the hinge-like closing of TOP (see also Figure 2).

**Figure 2 ijms-23-07297-f002:**
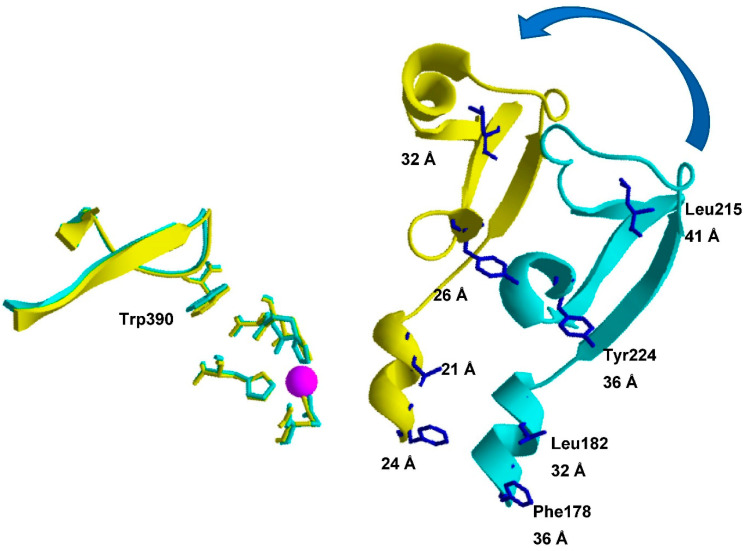
Positions of the D-A pairs (Trp390 and *p*NF residues) in the TOP structure. The Figure is a representation of a part of the overlay of the open and closed TOP molecules, fixing the zinc binding site HExxH + E + Zn^2+^ (pink sphere), shown in Figure 1C. The arrow highlights the relative change in position of domains D(onor) and A(cceptor) that should occur considering the hinge-like closing of TOP. Positions of the Trp390 residue (donor) and the fluorescence acceptor probe (*p*NF) are given together with the corresponding D-A distances. Figure was made by using Swiss pdb Viewer v.4.1.0 [42].

**Figure 3 ijms-23-07297-f003:**
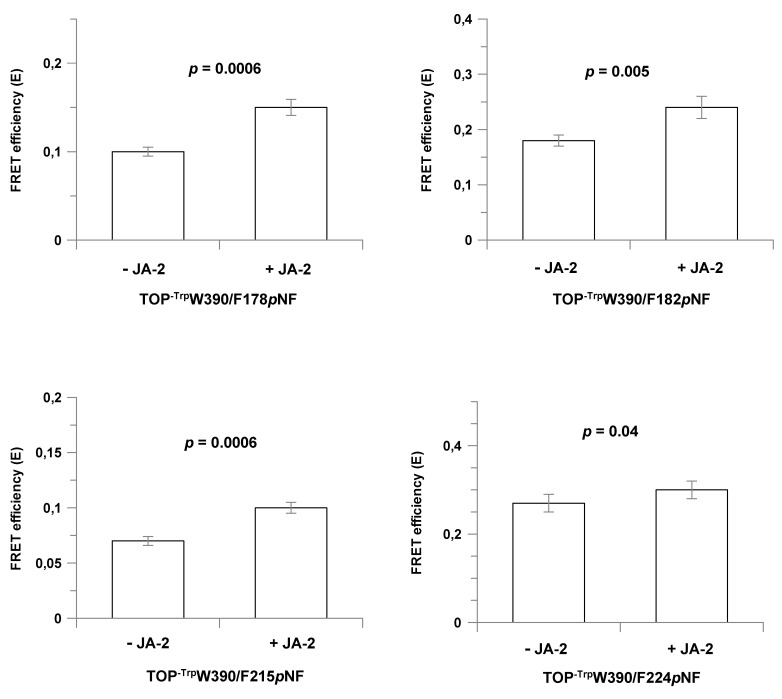
FRET efficiencies (E) obtained for the double labeled Trp-*p*NF TOP mutants in the absence or in the presence of JA-2. The *p-*values (*p*) were estimated by using the student’s *t* test. FRET efficiencies were determined at 37 °C in 50 mM Tris-HCl buffer, pH 7.4, containing 100 mM NaCl and 0.5 mM of DTT.

**Table 1 ijms-23-07297-t001:** Kinetic parameters of TOP proteins for the hydrolysis of substrate QFS and the inhibition constant *K*_i_ for JA-2.

Sample	*k*_cat_^a^ (s^−1^)	*K*_M_^a^ (µM)	*k*_cat_/*K*_M_^a^ (mM^−1^ s^−1^)	*K*_i_^a^ (µM)
WT TOP	0.4 ± 0.1	3.8 ± 0.3	105 ± 27	0.032 ± 0.001
TOP^-Trp^W390	0.039 ± 0.001	10.0 ± 0.5	3.9 ± 0.2	0.75 ± 0.04
TOP^-Trp^W390/F178*p*NF	0.0058 ± 0.0002	8.6 ± 0.7	0.67 ± 0.06	1.23 ± 0.08
TOP^-Trp^W390/L182*p*NF	0.0019 ± 0.0001	8.2 ± 0.8	0.23 ± 0.03	0.74 ± 0.06
TOP^-Trp^W390/L215*p*NF	0.042 ± 0.001	9.8 ± 0.4	4.3 ± 0.2	0.87 ± 0.05
TOP^-Trp^W390/Y224*p*NF	0.039 ± 0.002	10 ± 1	3.9 ± 0.4	1.16 ± 0.06

^a^ Kinetic parameters ± SD are calculated based on three to five assays. The kinetic parameters were determined at 37 °C in 50 mM Tris-HCl buffer, pH 7.4, containing 100 mM NaCl and 0.5 mM of DTT.

**Table 2 ijms-23-07297-t002:** FRET efficiencies (E) and the corresponding D-A distances of TOP mutants in absence and presence of the inhibitor JA-2. The expected D-A distances based on the “open” TOP crystal structure and a “closed” TOP structural model (Appendix A) are also presented as well as the fraction of the mutant in the closed state in the absence of JA-2.

FRET Mutants	Measured Values from Fluorescence Assays				Expected Values Based on 3D Structure				Closed/ Open
	−JA-2		+JA-2		TOP Open		TOP Closed		−JA-2
	FRET Efficiency ^a^	Distance ^b^ Å	FRET Efficiency ^a^	Distance ^b^ Å	FRET Efficiency	Distance ^c^ Å	FRET Efficiency	Distance ^c^ Å	% in Closed State ^d^
TOP^-Trp^W390/F178*p*NF	0.10 ± 0.005	23	0.150 ± 0.009	21	0.007	36	0.075	24	65
TOP^-Trp^W390/L182*p*NF	0.18 ± 0.01	20	0.24 ± 0.02	19	0.014	32	0.120	21	73
TOP^-Trp^W390/L215*p*NF	0.07 ± 0.004	24	0.10 ± 0.005	23	<0.005	41	0.014	32	68
TOP^-Trp^W390/Y224*p*NF	0.27 ± 0.01	19	0.30 ± 0.02	18	0.007	36	0.048	26	90

^a^ Measured FRET efficiency values (E ± SD) are bases on 3 to 6 assays. Measurements were performed at 37 °C in 50 mM Tris-HCl buffer, pH 7.4, containing 100 mM NaCl and 0.5 mM of DTT. ^b^ D-A distances are calculated using Equation (1). ^c^ Distance between Cβ of donor and Cβ of acceptor. ^d^ In the absence of JA-2 using the 2 state model as discussed in the text.

**Table 3 ijms-23-07297-t003:** Catalytic efficiencies (*k*_cat_/*K*_M_) for the hydrolysis of substrate QFS by TOP WT and TOP F390W at different concentrations of NaCl.

NaCl [M]	WT TOP	TOP^-Trp^W390
0	1.0 ± 0.1	1.0 ± 0.1
0.1	0.66 ± 0.08	0.60 ± 0.08
0.25	0.51 ± 0.07	0.42 ± 0.07
0.5	0.46 ± 0.05	0.37 ± 0.05
1	0.71 ± 0.08	0.6 ± 0.1
2.5	3.6 ± 0.4	4.9 ± 0.6

Relative values were calculated based on the *k*_cat_/*K*_M_ values, measured without addition of NaCl in the assay buffer. Values ± SD are calculated based on the results of three to five assays.

**Table 4 ijms-23-07297-t004:** FRET efficiencies (E) determined for the double labeled TOP^-Trp^W390/Y224*p*NF mutant at different concentrations of NaCl. The fraction of the mutant in the closed state, calculated assuming that TOP has only two possible conformations, open and closed, are also presented.

NaCl [M]	FRET Efficiency ^a^	% in Closed State ^b^
0	0.27 ± 0.02	90
0.1	0.26 ± 0.02	86
0.25	0.17 ± 0.01	55
0.5	0.11 ± 0.01	34
1	nd	-
2.5	0.35 ± 0.04	100

^a^ Measured FRET efficiency values (E ± SD) are based on three to six assays. ^b^ Calculated assuming that TOP has only two possible conformations, open and closed. nd = not determined.

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
