# Peer review of "Probing the Conformational States of Thimet Oligopeptidase in Solution"

_ijms, 2022, doi:10.3390/ijms23137297_

Round 1
Reviewer 1 Report
Protein conformational dynamic is an intrinsic property of proteins and plays important roles in biological functions. Marcelo et al conducted biochemical approaches to construct donor(Try390)-acceptor(pNF) pair at TOP and used inhibitor binding to probe the open-close conformation states that may be involved in the enzymatic catalysis. The research carried out by the authors is novel and the overall quality of the presentation is high. The work is suitable for publication in IJMS after addressing the following concerns.
Major points:
1. Controls are needed to ensure that the enhanced FRET efficiency from inhibitor binding is due to the conformational change induced the decrease of donor-acceptor distance rather than the simple explanation that JA-2 acts as an acceptor to Trp390.
2. Is the significantly reduced reactivity of Trp mutants due to the mutation driving the equilibrium towards the open conformation, given the authors argue the closed conformation of the enzyme has the higher activity than the open conformation?
Minor points:
1. Denote ‘MA’;
2. Provide the structure of JA-2;
3. The NaCl concentration used in Table 1, 2 and figure 3 need to be clarified.
Reviewer 2 Report
This manuscript describes the analysis of the conformational change of the protein thimet oligopeptidase (TOP) in solution. TOP variants containing site-specific mutations have been produced to measure the hinge bending between two domains using fluorescence (FRET) assays. Fluorescence measurements were executed in the presence and absence of a (competitive) inhibitor and at different NaCl concentrations. The results show that binding to the inhibitor induces an increase in the closed conformation, i.e. a change in hinge-bending motion and that the conformational state depends on the NaCl concentration. Although the hinge-bending movement has been shown/predicted in published experiments, including X-ray crystallography, molecular simulations and biochemical characterization, this is the first report on the measurement of a hinge-bending movement in metallopeptidases in solution. The outcome of the experiments in this manuscript does not lead to completely novel implications of metallopeptidase function and conformational changes, however the performed experiments are elegant, the results are sound and the data are well-presented. Therefore, the application described in this manuscript will be of interest to researchers exploring conformational changes within proteins in solution. I have a few comments regarding the experiments.
Major comments:
1) My main comment concerns the mutation of the Trp residues. The Authors have replaced each of the Trp residues to identify a Trp residue that could serve as FRET acceptor. Expression data show that Trp390 is essential for TOP solubility, which makes this residue the best candidate as acceptor. However, the TOP-TrpW390 variant – in which all other Trp residues have been replaced by Phe residues – shows strongly reduced activity, as presented by the kcat/Km (and Ki value).
- It would be nice to add a figure (e.g. Fig 2b) where the position of the Trp residues is indicated within the crystal structure.
- Have the individual Trp mutants (except Trp390Phe which appears insoluble) been tested for activity using the QFS substrate? This could indicate which of these Trp residues contributes most to the activity.
- The catalytic efficiency (kcat/Km) of the TOP-TrpW390 variant is about 27-fold reduced (Table 1). Is this (predominantly) due to a change in kcat or Km?
- Could the Authors comment on the potential effect of the Trp substitutions within the TOP-TrpW390 mutant on the conformational heterogeneity in solution compared to wt-TOP? Since activity of this variant is reduced, this could also be caused by a dis-balance in the open/intermediate/closed states and one could argue that this variant does not reflect the state of wt-TOP in solution.
2) Are there other molecules known that could modulate activity and the equilibrium of open vs closed state in vivo, i.e. cofactors, associated proteins?
3) An increase in NaCl concentration from 0 to 1.0 M reduces the activity of wt and TOP-TrpW390 and this appears to be correlated with the fraction of protein being in the closed state. This is also in line with the observation that at 2.5 M NaCl the activity is increased, since all protein molecules are within the closed state. Although I agree with the statement that the conformational state of the protein is correlated with activity, I was wondering if it is known (e.g. from literature) if the NaCl concentration - or ionic strength - also affects the physical state of the protein (aggregated, insoluble, oligomerized,..) which could contribute to a change in activity.
Minor comments:
1) Line 130: “Analysis of … “. Pleas add a reference to Table 1 where the data is shown and mention that activity is reduced: I would suggest to rephrase this sentence to: “Analysis of … showed that mutant TOP-TrpW390 was active, albeit significantly reduced when compared to WT TOP (Table 1)“.
Reviewer 3 Report
This paper can be accepted as it is.
Round 2
Reviewer 1 Report
The draft has been improved.